# Comparative performance of a commercial and in-house Mp1p antigen-detecting enzyme immunoassay for the rapid diagnosis of talaromycosis

Joseph Barwatt[1ʘ], Lottie Brown [iD][2,3ʘ*], Nguyen Thi Mai Thu[1ʘ], Paula Gonzalez[1], Sruthi Venugopalan[4], Heera Natesan Sambath[1], Ngo Thi Hoa[5,6,7], Jian-Piao Cai[8], Kwok-Yung Yuen[8,9], Jasper Fuk-Woo Chan[8,9], Vo Trieu Ly[10], Thuy Le[1,5*]

1 Division of Infectious Disease and International Health, Duke University School of Medicine, Durham, North Carolina, United States of America, 2 Institute of Infection and Immunity, City St George's University, School of Health & Medical Sciences, London, United Kingdom, 3 St George's Hospital, St George's Hospital NHS Foundation Trust, London, United Kingdom, 4 Department of Internal Medicine, University of Nevada Reno School of Medicine, Reno, Nevada, United States of America, 5 Tropical Medicine Research Center for Talaromycosis, Parasitology and Microbiology Research Unit, Centre for Biomedical Research, Pham Ngoc Thach University of Medicine, Ho Chi Minh City, Vietnam, 6 Oxford University Clinical Research Unit, Ho Chi Minh City, Vietnam, 7 Nuffield department of Medicine, University of Oxford, Oxford, United Kingdom, 8 State Key Laboratory of Emerging Infectious Diseases, Carol Yu Centre for Infection, Department of Microbiology, School of Clinical Medicine, Li Ka Shing Faculty of Medicine, The University of Hong Kong, Pokfulam, Hong Kong, 9 Hainan Medical University – The University of Hong Kong Joint Laboratory of Tropical Infectious Diseases, Hainan Medical University, Haikou, Hainan, China, 10 Hospital for Tropical Diseases, Ho Chi Minh City, Vietnam

ʘ Authors contributed equally
* thuy.le@duke.edu (TL), lotbrown@sgul.ac.uk, lottie.brown@stgeorges.nhs.uk (LB)

## Abstract

### Background

Several antigen-detection assays have been developed for the rapid diagnosis of talaromycosis, but their utility has been limited by a lack of commercial options. The aim of this study was to perform a head-to-head comparison of the performance of our in-house monoclonal antibody-based Mp1p antigen-detecting enzyme immunoassay (EIA) with its recently-developed commercial platform.

### Methods

In this diagnostic accuracy, retrospective, case-cohort study, we compared the sensitivity, specificity, positive likelihood ratio (LR+) and negative likelihood ratio (LR-) of the commercial Wantai Mp1p EIA *versus* our in-house Mp1p EIA on paired plasma and urine samples from 424 hospitalized adults with advanced HIV disease, including 224 cases of proven talaromycosis, where *Talaromyces marneffei* was isolated in culture of blood or other clinical specimens, and 200 controls diagnosed with a range of

author responses alongside final, published articles. The editorial history of this article is available here: https://doi.org/10.1371/journal.pntd.0013248

**Data availability statement:** In accordance with data sharing regulations in Vietnam, raw data have been deposited to Pham Ngoc Thach University and are available upon request (dhpnt@pnt.edu.vn).

**Funding:** This work was supported by the National Institute of Health (Grant Number: R01AI143409 and U01AI169358 to TL). The funders had no role in study design, data collection and analysis, decision to publish, or preparation of the manuscript.

**Competing interests:** I have read the journal's policy and the authors of this manuscript have the following competing interests: TL has received investigator-initiated research grant from Gilead Science. No other authors disclose any competing interestings.

other opportunistic infections. All participants were randomly selected from prospective cohorts recruited between 2011 and 2019 from five centers in Vietnam.

## Results

The sensitivity of the Wantai and in-house Mp1p EIAs were comparable in plasma (95.1% *vs* 92.4%, $P = 0.11$), in urine (91.5% *vs* 87.1% $P = 0.07$), and in combined testing of plasma and urine (96.4% *vs* 96.0%, $P = 1.00$), where talaromycosis was diagnosed based on the positivity of either specimen. The specificity of the Wantai and in-house Mp1p EIAs were consistently high in plasma, in urine, and in combined testing (93 – 97%). The Wantai and in-house Mp1p EIAs provided substantially higher sensitivity than blood culture detection (96.4% and 96.0% *vs* 78.6%, $P < 0.001$). For both EIAs, LR+ were greater than 10 and LR- were less than 0.1, which increases the confidence to rule in or rule out talaromycosis.

## Conclusions

The diagnostic performance of the Wantai Mp1p EIA was comparable to our validated in-house Mp1p EIA, and significantly more sensitive than blood culture, offering a standardized tool for the rapid diagnosis of talaromycosis.

## Author summary

Talaromycosis is an invasive fungal disease endemic in Southeast Asia, and has emerged as a leading opportunistic infection in patients with advanced HIV disease and other immunocompromising conditions. The current diagnosis relying on isolation of *Talaromyces marneffei* in culture is insensitive, takes a median of five days to result, and often requires invasive sampling. Antigen-detection tests have the potential to improve outcomes from talaromycosis by reducing the time-to-diagnosis, but their clinical utility is limited by a lack of commercial options. We previously developed an enzyme immunoassay (EIA) which detects the Mp1p antigen of *T. marneffei* in minimally-invasive clinical specimens like urine and plasma. Our Mp1p EIA demonstrates higher sensitivity than blood culture, and has been developed as a commercial platform. In this study, we evaluated the comparative diagnostic performance of the commercial Wantai Mp1p EIA (Wantai Biological Pharmacy Enterprise Co. Ltd, Beijing, China) and our in-house Mp1p EIA in 424 individuals with advanced HIV disease, including 224 cases of culture-proven talaromycosis and 200 randomly-selected controls without clinical and culture evidence of talaromycosis. The sensitivity and specificity of the Wantai and in-house Mp1p EIA were comparable in plasma (sensitivity: 95% *vs* 92%, specificity: 96% *vs* 93%), urine (sensitivity: 92% *vs* 87%, specificity: 95% *vs* 97%), and combined testing of plasma and urine (sensitivity: 96% *vs* 96%, specificity: 94% *vs* 93%). Both the Wantai and in-house Mp1p EIAs provided substantially higher sensitivity than blood culture detection

(96% and 96% *vs* 79%, respectively). The Wantai Mp1p EIA offers a standardized commercially-available tool for the rapid diagnosis of talaromycosis.

## Introduction

*Talaromyces marneffei*, formerly *Penicillium marneffei*, is a thermally dimorphic fungus which causes the invasive fungal disease talaromycosis primarily affecting the immunocompromised host. In Southeast Asia, talaromycosis is a leading cause of mortality among people with advanced HIV disease (AHD), surpassing other major opportunistic infections like cryptococcal meningitis [1–4]. An estimated 17,300 cases and 4900 deaths occur each year, primarily among individuals with AHD living in Southeast Asia, though incidence is rising among people with immunocompromising conditions other than HIV and in travelers to endemic regions [5,6]. In 2022, the World Health Organization (WHO) included *T. marneffei* on the Fungal Priority Pathogen List, recognizing there is an urgent need for research and development in diagnostics and treatment for talaromycosis [7].

A major barrier to improving outcomes from talaromycosis is our inability to make a timely and accurate diagnosis. The mortality doubles from 24% to 50% if the diagnosis is delayed and reaches 100% if the diagnosis is missed [8]. The current diagnosis of talaromycosis relies on isolation of *T. marneffei* in culture of blood and other clinical specimens. However, culture requires 5–28 days for identification and is only able to detect talaromycosis in the late stages of disease, when treatment is the least effective [3,9]. Even in the presence of disseminated disease, blood culture is negative in a third of cases, which necessitates invasive sampling of skin lesions, lymph nodes or bone marrow for diagnosis [9]. Rapid, non-invasive, and non-culture diagnostics are needed to facilitate the early initiation of antifungal therapy and improve patient outcomes.

We have developed a monoclonal antibody (MAb)-based antigen detection enzyme immunoassay (EIA) targeting the *T. marneffei* cell wall mannoprotein Mp1p [10]. Mp1p is an important virulence factor for *T. marneffei*, is immunogenic, and is secreted in abundance in the blood and urine of patients during infection, making it a useful target for immuno-diagnostics [11]. We have validated the Mp1p EIA in plasma and urine samples of 372 culture-positive cases and 517 controls, demonstrating higher sensitivity compared to blood culture (86.3% *vs* 72.8%, *P*<0.001), while reducing the time-to-diagnosis (6 hours *vs* median 6.6±3.0 days). We showed that combined testing of plasma and urine samples further improved sensitivity compared to testing plasma alone (88.8% *vs* 82.9%, *P*<0.001) [12]. A commercial version of our in-house Mp1p EIA using similar antibodies was developed by Wantai (Wantai Biological Pharmacy Enterprise Co. Ltd, Beijing, China) and approved for clinical use in China in 2019. Clinical evaluations of the Wantai Mp1p EIA are limited to two cohorts in China (n=283 and n=350) [13,14], and comparative analyses with our in-house Mp1p EIA have not been performed. While both EIAs use similar antibodies, there are some key differences in the protocols. The Wantai Mp1p EIA preloads and fixes polyclonal antibodies (PAbs) onto the plate using proprietary stabilized reagents, allowing for shelf-life stability of up to 12 months. In contrast, our in-house Mp1p EIA plates must be preloaded overnight or within a few days prior to use. The Wantai Mp1p EIA incorporates a co-incubation step, in which circulating Mp1p antigens in human samples interact simultaneously with the anti-Mp1p detection antibody. This approach reduces the total assay time from 6 to 2 hours by eliminating the sequential incubation steps required in our in-house Mp1p EIA. The aim of our study is to perform a robust head-to-head comparison of the clinical performance of the Wantai *versus* our in-house Mp1p EIAs in paired clinical samples, using culture-proven talaromycosis as the reference standard.

## Methods

### Ethics statement

This diagnostic evaluation utilized archived plasma and urine samples collected as a sub-study of the multi-center Itraconazole *versus* Amphotericin B for Penicilliosis (IVAP) randomized controlled trial (approval number 329/QD-BVBND) and a

single-center, prospective cohort study (approval number 816//QD-BVBND). The sub study was approved by the Vietnam Ministry of Health, all five study sites in Vietnam, and the Oxford Tropical Research Ethical Committee in the UK. All participants gave written informed consent for specimens to be stored and used in this research.

## Study design and populations

In this diagnostic accuracy, case-cohort study, cases were included from two cohorts of hospitalized adults (≥ 18 years) with AHD and culture-confirmed talaromycosis: [1] The Itraconazole *versus* Amphotericin B for Penicilliosis (IVAP) randomized control trial conducted at five referral hospitals across Vietnam between 2011 and 2017 [15], and [2] A prospective talaromycosis screening cohort study conducted at the Hospital for Tropical Diseases in Ho Chi Minh City, Vietnam between 2018 and 2019 [16]. The reference standard was isolation of *T. marneffei* in culture of blood or other clinical specimens (including skin lesions, lymph nodes and bone marrow) from participants with a clinical syndrome consistent with talaromycosis [17]. Controls were selected from the prospective talaromycosis screening cohort of hospitalized adults with AHD who were determined to have no evidence of talaromycosis clinically or by culture of blood or other specimens over a 6-month follow-up period. Cases and controls were randomly selected from batches of cases and controls based on availability of sample aliquot and volume without knowledge of clinical and laboratory characteristics of participants. Plasma was stored at -80°C and urine at -20°C until they were thawed once for testing.

## The in-house Mp1p EIA

The Mp1p EIA protocol has been reported previously [12]. In brief, immunoplates (Nunc, Roskilde, Denmark) were coated with rabbit Mp1p polyclonal antibodies (PAbs) at a concentration of 5 µg/mL overnight at 4°C and were further blocked in Tris-base with 0.2% gelatin and 0.25% casein at 37°C for 2 hours. Aliquots of 100 µL of undiluted plasma and urine samples were added to the coated wells and incubated at 37°C for 1 hour. The plate was then washed six times with 0.05% Tween in phosphate-buffered saline (PBS) (Sigma, St Louis, Missouri, USA). After washing, 100 µL of 1:1000 diluted mouse Mp1p MAbs conjugated with biotin was added and incubated at room temperature (25°C) for 30 minutes, followed by incubation with streptavidin-horseradish peroxidase (HRP) (Agilent-Dako, Santa Clara, California, USA) at room temperature (25°C) for 30 minutes. Tetramethylbenzidine (Invitrogen, Carlsbad, California, USA) was then added. The reaction was stopped after 10 minutes by the addition of 0.3 M sulfuric acid. Finally, the plate was examined in an enzyme-linked immunosorbent assay (ELISA) reader (ACTGene, Piscataway, New Jersey, USA) at wavelength reading at 450 nm.

## The commercial wantai Mp1p EIA

The Wantai Mp1p EIA was performed according to manufacturer's instructions. Briefly, biotin-conjugate (20 µL) was added to the microwell plate which was pre-coated with polyclonal Mp1p antibodies isolated from Mp1p-Ag immunized rabbits. Aliquots of 50 µL of undiluted plasma and urine samples were added to the coated wells and incubated at 37°C for 30 minutes. The plates were then washed five times with deionized water. After washing, 100µL of 5% HRP conjugate was added to each well and the plate was incubated for 30 minutes at 37°C. The plate was then washed five times with wash buffer. Chromogen Solution A (50 µL) and Chromogen Solution B (50 µL) were added to each well and incubated at 37°C for 15 minutes. The reaction was stopped by the addition of Stop Solution (50 µL) and the plate was examined in an ELISA reader (ACTGene, USA) with wavelength reading at 450 nm.

## Sample size estimates

Sample size calculations were based on the Hajian-Tilaki's method for diagnostic test studies [18], previous reported sensitivity and specificity of our in-house Mp1p EIA [12], and a case:control ratio of 1:1. A sample size of 189 cases and 185 controls would provide a power of 80% to detect a sensitivity of 86% (error d = 0.07) and a specificity of 94% (error

d = 0.04) with an α = 0.05 for the Mp1p EIA [19]. Our sample size of 224 cases and 200 controls ensures a power higher than 80% for the sensitivity and specificity estimates.

## Statistical analyses

Baseline characteristics of participants were described using frequency and proportions for continuous variables, and median and interquartile range (IQR) for categorical variables. Comparisons between categorical variables were performed using Chi-squared test, while comparisons between continuous variables were performed using the Wilcoxon rank-sum test. For the diagnostic analysis, receiver operating characteristic (ROC) curves were generated using Graph-Pad Prism 9.5.0 (GraphPad Software Inc., San Diego, California, USA). The assay cut-offs were determined based on the Youden index on the ROC curve, which maximizes true positives and minimizes false positives. For each assay, the discrimination power between cases and controls was determined by calculating the area under the ROC curve (AUC). Point estimates and 95% confidence intervals (CI) were calculated for sensitivity, specificity, positive likelihood ratios (LR+) and negative likelihood ratios (LR-). In a retrospective case-cohort study, where the disease prevalence is artificially generated, calculation of likelihood ratios is a more appropriate approach to interpret assay performance than predictive values which are prevalence-dependent and could be misleading. Sensitivity and specificity of the Wantai and in-house Mp1p EIA were compared using the McNemar's test for paired data. All statistical analyses were performed using R software version 4.2.2 (R Foundation for Statistical Computing, Vienna, Austria).

## Results

### Characteristics of study participants

Fig 1 shows the selection of the study participants and the alternative diagnoses of the controls based on the STARD guidelines [20]. A total of 224 cases of culture-confirmed talaromycosis were recruited, including all participants who met inclusion criteria from the prospective talaromycosis screening cohort study (n = 70) and a random selection (n = 154) of 440 eligible participants who enrolled in the IVAP clinical trial. Cases were diagnosed by positive culture of blood (n = 175, 78.1%), skin lesions (n = 141, 62.9%), lymph nodes (n = 10, 4.5%), and bone marrow (n = 2, 0.9%). Controls (n = 200) were selected at random from 452 participants who enrolled in the prospective screening cohort study and demonstrated no clinical or culture evidence of talaromycosis over a 6-month follow-up period. Controls were diagnosed with a range of other opportunistic infections including oral/esophageal candidiasis (n = 141, 70.5%), *Pneumocystis* pneumonia (n = 79, 39.5%), tuberculosis (n = 24, 12.0%), toxoplasmosis (n = 18, 9.0%) and cryptococcosis (n = 17, 8.5%). Cases and controls were similar in age (median age 33 years *vs* 34 years), *P* = 0.23. There were fewer males in cases compared to controls (67% *vs* 84%, *P* < 0.001), and more people who inject drugs (PWID) (25% *vs* 15%, *P* = 0.02). All participants had CD4 count below 200 cells/mL. Median CD4 count was lower in cases compared to controls (10 cells/mL *vs* 19 cells/mL), *P* = 0.001 (Table 1).

### Diagnostic performance of the wantai versus in-house Mp1p EIAs in plasma samples

Fig 2A displays the optical density (OD) distribution of cases and controls obtained by the Wantai *vs* in-house Mp1p EIAs in plasma samples. There was a significant difference in the OD distribution of cases *vs* controls for both assays (*P* < 0.001). Fig 2B shows the ROC curves and corresponding AUC values. The Wantai and in-house Mp1p EIAs demonstrated similarly excellent power in discriminating disease from no disease in plasma, with AUC of 96.0% (95% CI 93.9 – 98.1%) *vs* 96.5% (95% CI 94.7 – 98.3%), *P* = 0.66. Based on the Youden index, the optimal OD cut-off for the Wantai EIA in plasma were 0.09 (which is very close to the commercially-defined cut-off of 0.1) and for the in-house Mp1p EIA of 0.2. Based on these cut-off, the Wantai and in-house Mp1p EIAs demonstrated similar sensitivity of 95.1% (95% CI 91.4 – 97.5%) *vs* 92.4% (95% CI 88.1 – 95.5%), *P* = 0.11, and similar specificity of 95.5% (95% CI 91.6 – 97.9%) *vs* 93.0% (95% CI 88.5 – 96.1%), *P* = 0.18.

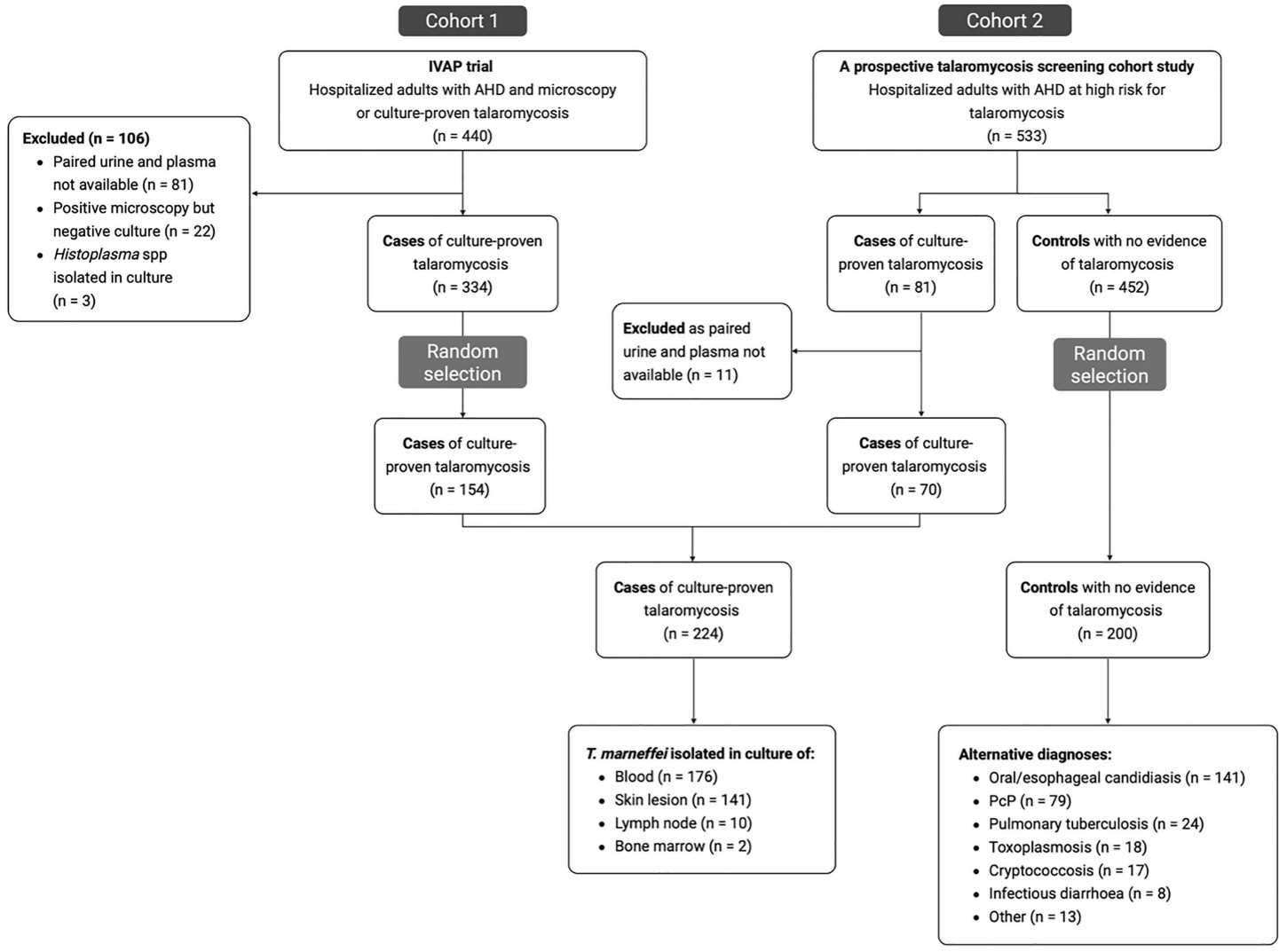

**Fig 1. Flow diagram of participant selection.** Cases included 154 participants from the Itraconazole *versus* Amphotericin for Penicillosis (IVAP) trial plus 70 participants from the prospective talaromycosis screening cohort study. All cases had culture-proven talaromycosis. Controls included 200 participants from the prospective talaromycosis screening study who demonstrated no clinical or culture evidence of talaromycosis over a 6-month follow up period. Paired plasma and urine specimens were available for all included participants. Cases and controls were randomly selected from batches of cases and controls based on availability of sample aliquot and volume without knowledge of clinical and laboratory characteristics of patients. Abbreviations: PcP, *Pneumocystis* pneumonia; spp, species; TB, tuberculosis.

The positive likelihood ratio (LR+) (i.e., probability of a person with the disease testing positive divided by the probability of a person without the disease testing positive) was 21.1 for the Wantai and 13.2 for the in-house Mp1p EIAs. Of note, a LR+ of 10 or greater indicates a large increase (+45% or higher) in the probability of disease with a positive test. The negative likelihood ratio (LR-) (i.e., the probability of a person with the disease testing negative divided by the probability of a person without the disease testing negative) was 0.05 for the Wantai *vs* 0.08 for the in-house Mp1p EIAs. Of note, a LR- of 0.1 or lower indicates a large decrease (-45% or higher) in the probability of no disease with a negative test (Table 2).

**Table 1. The demographic and laboratory characteristics of 224 participants with talaromycosis (cases) and 200 participants without talaromycosis (controls).**

| Characteristics | Overall n = 424 | Cases n = 224 | Controls n = 200 | *P* value |
|---|---|---|---|---|
| Age (years) | 34 (29 – 40) | 33 (28 – 38) | 34 (29 – 41) | 0.23 |
| Gender (male) | 318 (75) | 151 (67) | 167 (84) | < 0.001 |
| PWID (yes) | 85 (20) | 55 (25) | 30 (15) | 0.02 |
| CD4 count (cells/mL) | 13 (6 – 28)* | 10 (6 – 23) | 19 (7 – 36) | 0.001 |

Summary statistics are median (interquartile range, IQR) for continuous variables and frequency (%) for categorical variables. Chi-squared test was used for categorical variables and the Wilcoxon rank-sum test for continuous variables.

*CD4 count available for participants (n = 420).

Abbreviations: PWID, people who inject drugs.

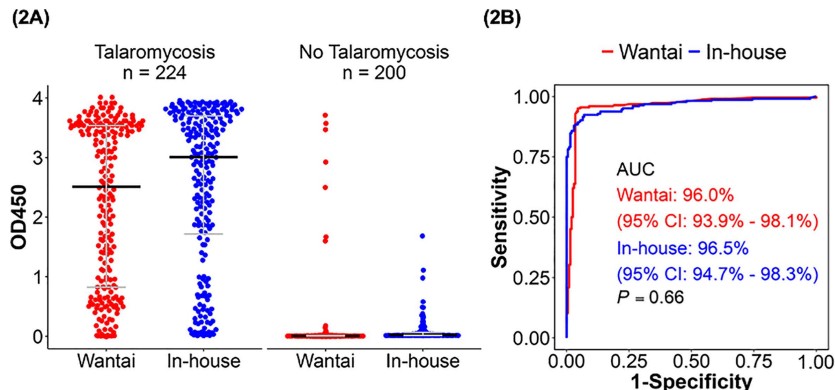

**Fig 2. The diagnostic performance of the Wantai *versus* in-house Mp1p enzyme immunoassays (EIA) in plasma. (2A)** The optical density (OD) distribution of 224 cases and 200 controls evaluated by the Wantai Mp1p EIA (red) *vs* the in-house Mp1p EIA (blue) in plasma. The median OD was significantly higher in cases compared with controls in both assays (*P*<0.001). **(2B)** Receiver Operative Curve (ROC) for the Wantai Mp1p EIA (red) *vs* the in-house Mp1p EIA (blue). Area under curve (AUC) of the Wantai Mp1p EIA is similar to in-house Mp1p EIA, 96.0% *vs* 96.5% (*P*=0.66) (DeLong's test). Abbreviations: 95% CI, 95% confidence interval; AUC, area under curve; OD450, optical density at a wavelength of 450 nanometers.

## Diagnostic performance of the wantai versus in-house Mp1p EIAs in urine samples

Fig 3A displays the OD distribution of cases and controls obtained by the Wantai and in-house Mp1p EIAs in urine. There were significant differences in the OD distribution for cases *vs* controls for both assays *(P*<0.001). Fig 3B shows the ROC curves and corresponding AUC values for each assay. Based on the Youden Index, the optimal OD cut-off for the Wantai in urine were 0.09 (which is very close to the commercially-defined cut-off of 0.1) and was 0.13 for the in-house Mp1p EIA. The Wantai Mp1p EIA had slightly lower power in discriminating between disease and no disease in urine compared to the in-house Mp1p EIA, with AUC of 92.5% (95% CI 89.6 – 95.4%) *vs* 95.4% (95% CI 93.3 –97.5%), but the difference did not reach statistical significance *P*=0.06. The Wantai and in-house Mp1p EIAs demonstrated similar sensitivity of 91.5% (95% CI 87.1 – 94.8%) *vs* 87.1% (95% CI 81.9 – 91.2%), *P*=0.07, and similar specificity of 94.5% (95% CI 90.4 – 97.2%) *vs* 97.0% (95% CI 93.6 – 98.9%), *P*=0.13. LR + was 16.6 for the Wantai Mp1p EIA and 29.0 for the in-house Mp1p EIA in urine, while LR- was 0.01 for the Wantai and 0.13 for the in-house Mp1p EIAs (Table 3).

**Table 2. Comparative diagnostic performance of the Wantai versus in-house Mp1p EIAs in plasma samples.**

| | Wantai EIA (OD ≥ 0.09) | | In-house EIA (OD ≥ 0.21) | | |
|---|---|---|---|---|---|
| | Cases | Controls | Cases | Controls | *P* values |
| Positive (n) | 213 (TP) | 9 (FP) | 207 (TP) | 14 (FP) | |
| Negative (n) | 11 (FN) | 191(TN) | 17 (FN) | 186 (TN) | |
| Sum (n) | 224 | 200 | 224 | 200 | |
| Sensitivity (%) | 213/224 95.1 (91.4 − 97.5) | | 207/224 92.4 (88.1 − 95.5) | | 0.11 |
| Specificity (%) | 191/200 95.5 (91.6 − 97.9) | | 186/200 93.0 (88.5 − 96.1) | | 0.18 |
| LR+ | 21.1 (11.2 − 40.0) | | 13.2 (8.0 − 21.9) | | 0.08 |
| LR- | 0.05 (0.03 − 0.09) | | 0.08 (0.05 − 0.13) | | 0.05 |

Data for diagnostic performance presented as point estimate with 95% confidence interval (CI). Statistical significance of the difference in sensitivity and specificity was assessed using McNemar's test, and in likelihood ratios using likelihood ratio test.

Abbreviations: TP, true positive; FP, false positive; FN, false negative; TN, true negative; LR +, positive likelihood ratio; LR-, negative likelihood ratio.

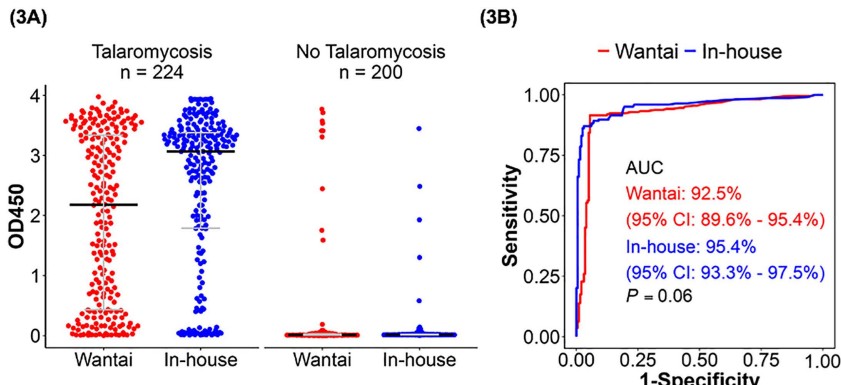

**Fig 3. The diagnostic performance of the Wantai and in-house Mp1p EIA in urine. (3A)** The optical density (OD) distribution of 224 cases and 200 controls from the Wantai (red) and the in-house Mp1p EIA (blue) on plasma. The median OD was significantly higher in cases compared with controls in both assays (*P*<0.001). **(3B)** Receiver Operative Curve (ROC) for the Wantai Mp1p EIA (red) and the in-house Mp1p EIA (blue). Test accuracy or area under the curve (AUC) of the Wantai Mp1p EIA is similar to the in-house Mp1p EIA, 92.5% *vs* 95.4% (*P*=0.06) (DeLong's test). Abbreviations: 95% CI, 95% confidence interval; AUC, area under curve; OD450, optical density at a wavelength of 450 nanometers.

## Diagnostic performance of the wantai versus in-house EIAs in combined testing of plasma and urine samples

Table 4 shows the comparative performance of the Wantai and in-house Mp1p EIAs in paired plasma and urine samples for detecting cases of culture-confirmed talaromycosis. For the combined testing of plasma and urine, where talaromycosis was diagnosed based on positivity of either plasma or urine, sensitivity increased from 91.5% and 95.1% to 96.4% (95% CI 93.1 − 98.4%) for the Wantai Mp1p EIA and from 87.1% and 92.4% to 96.0% (92.5 − 98.1%) for the in-house Mp1p EIA (*P*=1.0). Both assays correctly detected 213 out of 224 cases (95.1%) of talaromycosis in paired plasma and urine samples. An additional three cases were detected by the Wantai Mp1p EIA and two by the in-house Mp1p EIA (Fig 4).

**Table 3. Comparative diagnostic performance of the Wantai and in-house Mp1p EIAs in urine samples.**

| | Wantai (OD ≥ 0.09) | | In-house EIA (OD ≥ 0.13) | | |
|---|---|---|---|---|---|
| | Cases | Controls | Cases | Controls | *P* values |
| Positive | 205 (TP) | 11(FP) | 195 (TP) | 6 (FP) | |
| Negative | 19 (FN) | 189 (TN) | 29 (FN) | 194 (TN) | |
| Sum | 224 | 200 | 224 | 200 | |
| Sensitivity (%) | 205/224 91.5 (87.1 – 94.8) | | 195/224 87.1 (81.9 – 91.2) | | 0.07 |
| Specificity (%) | 189/200 94.5 (90.4 – 97.2) | | 194/200 97.0 (93.6 – 98.9) | | 0.13 |
| LR+ | 16.6 (9.4 – 29.6) | | 29.0 (13.2 – 63.9) | | 0.09 |
| LR- | 0.09 (0.06 – 0.14) | | 0.13 (0.10 – 0.19) | | 0.09 |

Data for diagnostic performance presented as point estimate with 95% confidence interval (CI). Statistical significance of the difference in sensitivity and specificity was assessed using McNemar's test, and in likelihood ratios using likelihood ratio test.

Abbreviations: TP, true positive; FP, false positive; FN, false negative; TN, true negative; LR +, positive likelihood ratio; LR-, negative likelihood ratio.

**Table 4. Comparative performance of the Wantai and in-house Mp1p EIAs in combined urine and plasma samples.**

| | Wantai | | In-house EIA | | |
|---|---|---|---|---|---|
| | Cases | Controls | Cases | Controls | *P* values |
| Positive (n) | 216 (TP) | 13(FP) | 215 (TP) | 15 (FP) | |
| Negative (n) | 8 (FN) | 187 (TN) | 9 (FN) | 185 (TN) | |
| Sum (n) | 224 | 200 | 224 | 200 | |
| Sensitivity (%) | 216/224 96.4 (93.1 – 98.4) | | 215/224 96.0 (92.5 – 98.1) | | 1.00 |
| Specificity (%) | 187/200 93.5 (89.1 – 96.5) | | 185/200 92.5 (87.9 – 95.7) | | 0.75 |
| LR+ | 14.8 (8.8 – 25.1) | | 12.8 (7.9 – 20.8) | | 0.74 |
| LR- | 0.04 (0.02 – 0.08) | | 0.04 (0.02 – 0.08) | | 0.74 |

Data for diagnostic performance presented as point estimate with 95% confidence interval (CI). Statistical significance of the difference in sensitivity and specificity was assessed using McNemar's test, and in likelihood ratios using likelihood ratio test.

Abbreviations: TP, true positive; FP, false positive; FN, false negative; TN, true negative; LR +, positive likelihood ratio; LR-, negative likelihood ratio.

## Sensitivity of the wantai and in-house Mp1p EIA versus blood culture detection

Combined testing of urine and plasma by the Wantai and in-house Mp1p EIAs provided substantially higher sensitivity than blood culture detection (96.4% and 96.0% *vs* 78.6%, *P*<0.001, Table 5). Testing of urine or plasma alone also provided significantly higher sensitivity than blood culture. In participants with positive blood culture, the sensitivity of the Wantai and in-house Mp1p EIA was 96.0% (95% CI: 91.7 – 98.2%) and 96.6% (95% CI: 92.4 – 98.6%), respectively, *P*=1.0. In participants with negative blood culture, sensitivity was similar for the Wantai Mp1p EIA (97.9%, 95% CI: 87.5 – 99.9%) *vs* the in-house Mp1p EIA (93.8%, 95% CI: 81.8 – 98.4%), *P*=0.48 (Table 5).

## Characteristics of false negatives and false positives

Of the six false negative cases (where antigen was negative by both the Wantai and in-house Mp1p EIA in both plasma and urine samples), five were diagnosed by positive blood culture and one by positive skin lesion culture. Of the nine false

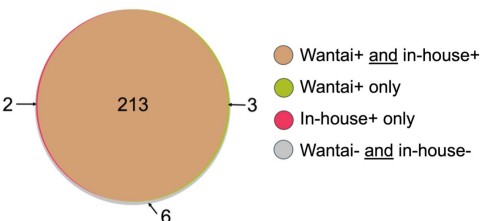

**Fig 4. Venn diagram demonstrating the diagnostic yield of the Wantai and in-house Mp1p EIA on paired plasma and urine samples.** The Wantai EIA detected 96.4% of cases (216/224) and the in-house Mp1p EIA detected 96.0% of cases of talaromycosis. Abbreviations: +, positive; -, negative.

**Table 5. Comparative diagnostic performance of the Wantai Mp1p EIA and in-house Mp1p EIA *versus* blood culture.**

| | Blood culture | Wantai Mp1p EIA | In-house Mp1p EIA | *P* value |
|---|---|---|---|---|
| Blood culture-positive cases (n = 176) | 176 (100%) | 169 (96.0%, 95% CI 91.7 – 98.2) | 170 (96.6%, 95% CI 92.4 – 98.6) | 1 (Wantai *vs* In-house) |
| Blood culture-negative cases (n = 48) | – | 47 (97.9%, 95% CI 87.5 - 99.9) | 45 (93.8%, 95% CI 81.8 - 98.4) | 0.48 (Wantai *vs* In-house) |
| Total cases (n = 224) | 176 (78.6%, 95% CI 72.5 - 83.6) | 216 (96.4%, 95% CI 92.8 – 98.3) | 215 (96.0%, 95% CI 92.3 – 98.0) | < 0.001 (Blood culture *vs* Wantai or in-house) |

Notes: Statistical significance of the difference in sensitivity between two groups was assessed using McNemar's test.

Abbreviations: 95% CI, 95% confidence interval; EIA, enzyme immunoassay.

positive cases, three were diagnosed with oral/esophageal candidiasis, while the remaining six cases were diagnosed with *Pneumocystis* pneumonia, toxoplasmosis, and infectious diarrhea.

## Discussion

Prompt diagnosis and treatment are critical to reducing the high mortality rate of talaromycosis (up to 30%) [5]. Currently, most cases of talaromycosis are diagnosed by culture, which is slow and insensitive. Important advances in the development of rapid diagnostics have been made in recent years, with antigen-detection tests like the Mp1p EIA demonstrating superior sensitivity and excellent specificity compared to blood culture and significantly reducing the time to diagnosis and initiation of antifungal therapy [12]. However, the utility of these antigen tests has been limited by a lack of commercial options. There is a clear need for reliable, affordable, and widely-available commercial antigen-detection tests to improve the diagnosis of talaromycosis. In this multi-center, case-cohort study of over 400 participants living with AHD in Vietnam, we evaluated the sensitivity and specificity of the commercial Wantai Mp1p EIAs and demonstrated comparable performance with our extensively validated in-house Mp1p EIA. The sensitivity of both EIAs improved further when performing combined testing of plasma and urine (from 87 – 95% to 96%), and this is our recommended approach to antigen testing. However, where it is not possible or practical to obtain both specimen types, the sensitivity of either urine or plasma in both EIAs was still substantially higher than blood culture. The specificity of the Wantai and the in-house EIAs was similarly high in plasma, urine, and combined plasma and urine testing (93 – 97%). Both the Wantai and the in-house Mp1p EIAs demonstrated high LR+ (> [10]) and low LR- (<0.1), suggesting promising utility as rapid, rule-in and rule-out tests for talaromycosis in people with AHD in Vietnam.

There are several possible causes for the false positives and false negatives observed in our study. Heterophilic antibody interference may have led to false positives due to non-specific binding of human antibodies with the rabbit-derived Mp1p antibodies. While previous studies do not report cross-reactivity of the Mp1p EIA with other major

pathogenic fungi, including *Cryptococcus, Candida, Histoplasma,* and *Aspergillus*, there remains the potential for cross-reactivity with less common fungi [12]. The baseline levels of circulating Mp1p in populations residing in endemic areas is unknown. Environmental exposure could lead to detectable antigenemia in the absence of invasive disease. Further studies are needed to define antigen thresholds that distinguish true infection from background exposure in high-burden settings. We did not suspect low fungal burden as a cause of false negatives in our study, as five of the six cases were positive by blood culture. False negatives may have occurred in people with prior exposure to *T. marneffei,* due to sequestration of Mp1p antigen to circulating antibodies, forming immune complexes and reducing the levels of free Mp1p. Further investigation is needed to characterize the adaptive immune response to *T. marneffei.* Variations in the Mp1p epitope may have arisen as a result of genetic diversity across different strains of *T. marneffei*, potentially contributing to false negative results [21].

The Wantai Mp1p EIA has previously been evaluated in two studies from Guangzhou, China [13,14]. These single-center, retrospective, cross-sectional studies examined the diagnostic performance of Mp1p EIA against culture, in a combined total of 633 participants with AHD and suspected talaromycosis. While the high specificity (97%) of the Wantai Mp1p EIA in these two studies was consistent with our study from Vietnam, the sensitivity was substantially lower (72 – 77%). Possible reasons for the difference in sensitivity are the different specimen types used for EIA analysis (combined plasma/urine in our study *versus* serum in the two studies from China), and differences in the study populations, with the Chinese studies including specimens from participants who had already been initiated on antifungal therapy, which may compromise sensitivity. Another possible explanation is that assay performance may be clade-specific. Genomic studies of *T. marneffei* isolates have revealed significant genetic variations and distinct geographical clades, which could impact detection by antigen tests, though this remains understudied and is an area of research need [21].

In addition to the rapid diagnosis of talaromycosis, Mp1p antigen detection may have a role in screening, prognostication, and monitoring of treatment response. *T. marneffei* antigenemia (detected by our in-house Mp1p EIA) has been shown to precede blood culture positivity by up to 16 weeks [22]. Like the serum Cryptococcal Antigen test (CrAg) for cryptococcal meningitis, antigen tests for talaromycosis have the potential be used for targeted screening of high-risk patients, such as people with AHD and CD4 ≤ 200 cells/mL, to facilitate early treatment before the onset of symptoms [16]. Mp1p antigen testing could also be used to monitor fungal clearance and response to treatment, through quantification of circulating Mp1p antigens determined by optical density readers for ELISA. Further, prospective trials are needed to determine the value of Mp1p antigen testing for prognostication and screen and treat strategies, including the evaluation of commercial kits.

There are some limitations to our study. Firstly, our evaluation was performed on archived clinical specimens collected between 2012 and 2019. Protein in older samples may have degraded, which could lead to an underestimation of assay sensitivity. Secondly, there were some significant differences in the characteristics of the cases and controls including the proportion of male participants, the prevalence of PWID and the median CD4 cell count. The difference in median CD4 count (10 cells/mm$^3$ for cases *vs* 19 cells/mm$^3$ for controls) is not clinically significant since both are considered severely immunocompromised with the same risk of opportunistic infections. As a case-cohort study, our control population consisted of clinically relevant participants with AHD and other opportunistic infections, which are representative of the population at risk and being tested for talaromycosis. Control participants were followed up for a minimum of 6 months and monitored for the development of talaromycosis, which minimized the risk of misclassification of controls. Finally, our evaluation was limited to individuals with AHD living in Vietnam. Given the evolving epidemiology of talaromycosis, future evaluations should include individuals with immunocompromising disorders other than HIV and in migrants or travelers from endemic regions. Our findings should be validated in larger, prospective cohort studies of individuals with and without AHD, living across a range of endemic and non-endemic regions.

Currently, the Wantai Mp1p EIA kit is only available in China, and the company is transitioning to a lateral flow assay (LFA) platform. The Mp1p EIA is also being developed as a commercial LFA by IMMY (IMMY diagnostics, Oklahoma, USA) showing comparable performance as the Mp1p EIA and is undergoing prospective clinical validation [23]. While EIA-based diagnostics require laboratory infrastructure and trained personnel, LFA testing is simple to perform with no equipment and is particularly well suited for point-of-care diagnosis in low resource and community settings. While the specificity remains high (98%), the sensitivity of the Mp1p LFA is slightly reduced compared to the Mp1p EIA (91% *versus* 95% in a single study of 292 adults with AHD) and the analytical limit of detection is lower [23], which is expected of LFA. Several other antigen-detection assays are being actively developed. These include the 4D1 EIA and its point-of-care LFA [24,25], and the Mp1p D4 POCT [26], a self-contained immunoassay platform which integrates all reagents within a capillary-driven passive microfluidic cassette to minimize user intervention and withstand extreme environment conditions. For neglected tropical diseases like talaromycosis, collaborations and partnerships between healthcare professionals, researchers, industry partners, policy makers and other stakeholders are necessary to continue the advancements of novel diagnostics, and to ensure sustained access to commercial options. Finally, we advocate for the WHO to recognize talaromycosis as a neglected tropical disease, as this could serve as a catalyst for action and drive progress in research and development for talaromycosis.

## Conclusion

The diagnostic performance of the commercial Wantai Mp1p EIA was comparable to our extensively validated in-house Mp1p EIA, and substantially more sensitive than blood culture for the diagnosis of talaromycosis. The Mp1p EIA increases the confidence to rapidly rule-in and rule-out talaromycosis in people with AHD. United efforts between researchers, industry partners and other stakeholders are needed to improve access to rapid diagnostics, including commercial options, for neglected diseases like talaromycosis.

## Acknowledgments

The commercial Mp1p EIA kits were provided in kind by Wantai Biological Pharmacy Enterprise Co. Ltd, Beijing, China.

## Author contributions

**Conceptualization:** Thuy Le.

**Data curation:** Joseph Barwatt, Nguyen Thi Mai Thu, Paula Gonzalez, Sruthi Venugopalan.

**Formal analysis:** Joseph Barwatt, Lottie Brown, Nguyen Thi Mai Thu, Sruthi Venugopalan, Heera Natesan Sambath.

**Funding acquisition:** Thuy Le.

**Investigation:** Joseph Barwatt, Nguyen Thi Mai Thu, Paula Gonzalez, Sruthi Venugopalan, Thuy Le.

**Methodology:** Nguyen Thi Mai Thu, Ngo Thi Hoa, Jian-Piao Cai, Kwok-Yung Yuen, Jasper Fuk-Woo Chan, Thuy Le.

**Project administration:** Joseph Barwatt, Lottie Brown, Paula Gonzalez, Sruthi Venugopalan.

**Resources:** Vo Trieu Ly, Thuy Le.

**Supervision:** Nguyen Thi Mai Thu, Ngo Thi Hoa, Jian-Piao Cai, Kwok-Yung Yuen, Jasper Fuk-Woo Chan, Vo Trieu Ly, Thuy Le.

**Visualization:** Joseph Barwatt, Lottie Brown, Nguyen Thi Mai Thu, Paula Gonzalez, Sruthi Venugopalan.

**Writing – original draft:** Joseph Barwatt, Lottie Brown, Nguyen Thi Mai Thu, Paula Gonzalez, Sruthi Venugopalan, Jasper Fuk-Woo Chan.

**Writing – review & editing:** Lottie Brown, Heera Natesan Sambath, Ngo Thi Hoa, Jian-Piao Cai, Kwok-Yung Yuen, Vo Trieu Ly, Thuy Le.

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
