## [Decision Letter · Decision Letter 0]

4 Jun 2025

PNTD-D-25-00725

Comparative Performance of a Commercial and In-House Mp1p antigen-detecting Enzyme Immunoassay for the Rapid Diagnosis of Talaromycosis

Dear Dr. Brown,

(This is great work Lottie- thank you for choosing Plos NTD. Regarding one point raised in review, I do favor keeping the "neglected" as we have including Talaro as such at this journal! Best, Josh)

Thank you for submitting your manuscript to PLOS Neglected Tropical Diseases. After careful consideration, we feel that it has merit but does not fully meet PLOS Neglected Tropical Diseases's publication criteria as it currently stands. Therefore, we invite you to submit a revised version of the manuscript that addresses the points raised during the review process.

Please submit your revised manuscript within 60 days Aug 03 2025 11:59PM. If you will need more time than this to complete your revisions, please reply to this message or contact the journal office at plosntds@plos.org. Please include the following items when submitting your revised manuscript:

We look forward to receiving your revised manuscript.

Kind regards,

Joshua Nosanchuk, MD

Section Editor

Shaden Kamhawi

co-Editor-in-Chief

Paul Brindley

co-Editor-in-Chief

**Journal Requirements:**

At this stage, the following Authors/Authors require contributions: Lottie Brown, Sruthi Venugopalan, Ngo Thi Hoa, Jian-Piao Cai, Kwok-Yung Yuen, Jasper F. W. Chan, and Vo Trieu Ly. Please ensure that the full contributions of each author are acknowledged in the "Add/Edit/Remove Authors" section of our submission form.

3) We note that your Data Availability Statement is currently as follows: "All relevant data are within the manuscript and its Supporting Information files.". Please confirm at this time whether or not your submission contains all raw data required to replicate the results of your study. Authors must share the “minimal data set” for their submission. PLOS defines the minimal data set to consist of the data required to replicate all study findings reported in the article, as well as related metadata and methods (https://journals.plos.org/plosone/s/data-availability#loc-minimal-data-set-definition).

- The points extracted from images for analysis..

**Reviewers' Comments:**

Reviewer's Responses to Questions

**Key Review Criteria Required for Acceptance?**

**Methods**

-Are the objectives of the study clearly articulated with a clear testable hypothesis stated?

-Is the study design appropriate to address the stated objectives?

-Is the population clearly described and appropriate for the hypothesis being tested?

-Is the sample size sufficient to ensure adequate power to address the hypothesis being tested?

-Were correct statistical analysis used to support conclusions?

-Are there concerns about ethical or regulatory requirements being met?

Reviewer #1: Lines 137-139: Consider defining other specimens were considered clinical

Lines 139: Consider referencing another paper that describes clinical syndromes consistent with talaro for those readers who may not be as familiar with the disease

Lines 139-143: Consider describing a bit more about how the controls were randomly selected? Numbering system? Same with the cases.

Lines 186-188: The authors may want to add a bit of reasoning into why likelihood ratios were chosen over the more commonly used predictive values. And a bit more commentary on how to use likelihood ratios.

Reviewer #2: (No Response)

**Results**

-Does the analysis presented match the analysis plan?

-Are the results clearly and completely presented?

-Are the figures (Tables, Images) of sufficient quality for clarity?

Reviewer #1: Lines 194-195: Consider removing this sentence and just citing figure after the next sentence for flow of manuscript. Consider doing the same for subsequent sections in the results sections.

Reviewer #2: (No Response)

**Conclusions**

-Are the conclusions supported by the data presented?

-Are the limitations of analysis clearly described?

-Do the authors discuss how these data can be helpful to advance our understanding of the topic under study?

-Is public health relevance addressed?

Reviewer #1: Lines 270-271: Consider adding reference to end of sentence.

Lines 342-343: Consider removing the designation as an NTD since technically not considered as an NTD by WHO; although, it very likely should be.

Reviewer #2: (No Response)

**Editorial and Data Presentation Modifications?**

Reviewer #1: Figure 1: Consider adding a bit more detail about randomization process.

Figure 4: Will ultimately defer to authors, but the value added from figure 4 may be minimal and could be removed or in supplemental material.

Reviewer #2: (No Response)

**Summary and General Comments**

Reviewer #1: - Lines 87-88: While vast majority of talaro occurs in those immunocompromised, disease in immunocompetent hosts have been documented. Consider modifying the end of the sentence to reflect this.

- Lines 93-96: Since this is PLoS NTDs, maybe consider including a sentence after WHO FPPL about how it is absent on WHOs NTD list for advocacy purposes? Will leave up to authors, but just a thought if authors think it will add to their report.

Lines 97-106: May want to include a bit of commentary about other promising point of care tests that the authors have worked on, but not commercially available yet.

Reviewer #2: The original manuscript by Barwatt and colleagues, which presents a comparative analysis of the diagnostic performance between a commercial Wantai EIA and an in-house EIA based on the Mp1p antigen of Talaromyces marneffei, is scientifically written, clear, concise, and focused. However, there remain several aspects that warrant further consideration, as outlined below:

1. Clarification of the Study’s Primary Objective:

It is important for the authors to more clearly articulate the main objective of this experiment. Given that both ELISA systems were developed based primarily on rabbit polyclonal antibodies against Mp1p, the resulting outcomes should reasonably align (as the authors have demonstrated), and significant discrepancies would not be expected. Therefore, the rationale and specific aims of the study should be explicitly stated.

2. Experimental Methodology:

There is some ambiguity regarding the experimental procedures. The methods used for the Wantai EIA and the in-house Mp1p EIA appear to differ slightly in their details. To enhance clarity and accessibility for the reader, the authors are encouraged to provide a schematic diagram or flow chart that visually summarizes the experimental workflow, rather than relying solely on textual descriptions.

3. False Negative and False Positive Characteristics:

Under the section addressing the characteristics of false negatives and false positives, the manuscript would benefit from additional detail. The occurrence of false negatives is not entirely unexpected, given that the mechanisms underlying the secretion of mannoprotein antigens in the pathogenesis of T. marneffei are not yet fully understood. The authors may wish to interpret this within the framework of the damage-response model of T. marneffei pathogenesis. Furthermore, baseline antigen levels within populations residing in endemic areas remain to be thoroughly investigated.

In contrast, the issue of false positives requires a more comprehensive discussion. In particular, potential cross-reactivity involving the rabbit anti-Mp1p antibodies should be considered and addressed in greater depth.

PLOS authors have the option to publish the peer review history of their article (what does this mean?). If published, this will include your full peer review and any attached files.

Reviewer #1: **Yes: **Dallas Smith

Reviewer #2: No

**Figure resubmission:**
---

## [Editor Report · Decision Letter 1]

16 Jun 2025

Dear Dr Brown,

Thank you, Lottie et al, for your thoughtful, rigorous, and complete answers to the comments on the original version of your work. We are pleased to inform you that your manuscript 'Comparative Performance of a Commercial and In-House Mp1p antigen-detecting Enzyme Immunoassay for the Rapid Diagnosis of Talaromycosis' has been provisionally accepted for publication in PLOS Neglected Tropical Diseases.

Best regards,

Joshua Nosanchuk, MD

Section Editor

Shaden Kamhawi

co-Editor-in-Chief

Paul Brindley

co-Editor-in-Chief

---

## [Editor Report · Acceptance letter]

Dear Dr Brown,

We are delighted to inform you that your manuscript, "Comparative Performance of a Commercial and In-House Mp1p antigen-detecting Enzyme Immunoassay for the Rapid Diagnosis of Talaromycosis," has been formally accepted for publication in PLOS Neglected Tropical Diseases.

Best regards,

Shaden Kamhawi

co-Editor-in-Chief

Paul Brindley

co-Editor-in-Chief
